# Conformalized Credal Regions for Classification with Ambiguous Ground Truth

**Michele Caprio**  *michele.caprio@manchester.ac.uk*
*Department of Computer Science*
*University of Manchester*

**David Stutz**  *dstutz@deepmind.com*
*DeepMind*

**Shuo Li**  *lishuo1@seas.upenn.edu*
*Department of Computer and Information Science*
*University of Pennsylvania*

**Arnaud Doucet**  *arnauddoucet@google.com*
*DeepMind*
*Department of Statistics, University of Oxford*

**Reviewed on OpenReview:** *https://openreview.net/forum?id=L7sQ8CW2FY*

## Abstract

An open question in *Imprecise Probabilistic Machine Learning* is how to empirically derive a credal region (i.e., a closed and convex family of probability measures on the output space) from the available data, without any prior knowledge or assumption. In classification problems, credal regions are a tool that is able to provide provable guarantees under realistic assumptions by characterizing the uncertainty about the distribution of the labels. Building on previous work, we show that credal regions can be directly constructed using conformal methods. This allows us to provide a novel extension of classical conformal prediction to problems with ambiguous ground truth, that is, when the exact labels for given inputs are not exactly known. The resulting construction enjoys desirable practical and theoretical properties: (i) conformal coverage guarantees, (ii) smaller prediction sets (compared to classical conformal prediction regions) and (iii) disentanglement of uncertainty sources (epistemic, aleatoric). We empirically verify our findings on both synthetic and real datasets.

## 1 Introduction

In most real-world applications of machine learning, especially in the context of safety-critical applications such as healthcare, researchers have found it difficult to reason with *precise* probabilities. Instead, practitioners have become comfortable reasoning in terms of families of probabilities, often in the form of closed and convex sets called *credal regions*. Imprecise probabilistic machine learning (IPML) (Denoeux, 2000; Zaffalon, 2002; Destercke et al., 2008; Caprio & Gong, 2023; Caprio & Mukherjee, 2023a;b; Dutta et al., 2023; Lu et al., 2024; Caprio, 2024; Caprio et al., 2024a;c;d; Sale et al., 2024) aims to develop machine learning theory and methods that work with such *imprecise* probabilities. Such tools allow to better quantify and disentangle different types of uncertainty, e.g., epistemic (model) and aleatoric (data) uncertainties, which play key roles in any machine learning system. However, other sources of uncertainty are also highly relevant; for example, uncertainty originating from the annotation process used to derive ground truth labels (Stutz et al., 2023a).

A crucial line of research in IPML is that of empirically deriving credal regions without any prior knowledge or assumption. The first steps in this direction were made recently by Cella & Martin (2022a;b). They discovered that, subject to a so-called *consonance* assumption (see Section 3.2) and given exchangeable

calibration data, the conformal transducer assigning a *p*-value to each possible label uniquely identifies a credal region. This is a very promising result, since this only relies on having to select a non-conformity measure, and the obtained coverage guarantee is valid irrespective of this choice. Unfortunately, Cella & Martin (2022a;b) do not provide an implementation of their results on credal regions in real-world, complex datasets. Moreover, they ignore the particularly interesting case of *ambiguous ground truth* (Stutz et al., 2023a;b) where labels are not crisp, but subject to uncertainty due to rater disagreement (Yan et al., 2014; Zheng et al., 2017) and imperfect labeling tools. In fact, calibration data are often not exactly classified; we can see this in machine learning (Dawid & Skene, 1979; Smyth et al., 1994), but especially in medicine (Feinstein & Cicchetti, 1990; McHugh, 2012; Raghu et al., 2019; Schaekermann, 2020), natural language processing (Pavlick & Kwiatkowski, 2019; Nie et al., 2020; Baan et al., 2022; Plank, 2022), and computer vision (Peterson et al., 2019a).

**Contributions.** In this paper, we follow an alternative way of constructing credal regions in a conformal way, inspired by recent work (Javanmardi et al., 2023; 2024; Lienen et al., 2023; Martin Bordini et al., 2023). Specifically, we directly construct conformalized families of probabilities and show that these are in fact credal regions, i.e. convex and closed. Furthermore, we extend this approach to classification problems with ambiguous ground truth, first studied in the context of conformal prediction by Stutz et al. (2023b;c).[1] These problems are extremely relevant from an applied point of view. Instead of a "precise" calibration set $(\mathbf{x^n}, \mathbf{y^n}) = \{(x_1, y_1), \ldots, (x_n, y_n)\} \subset \mathcal{X} \times \mathcal{Y}$,[2] we consider a calibration set $D_c = \{(x_i, \lambda_i)\}$ that encompasses the ground truth ambiguity, where the $\lambda_i$'s are probability vectors, and the $k$-th entry $\lambda_{i,k}$ is the probability of $k$ being the correct label for $x_i$.[3] To see that this is a generalization of the usual classification problems, notice that we can write a pair $(x_i, y_i)$ as a pair $(x_i, \lambda_i)$, where $\lambda_i$ is a one-hot encoding vector with entry 1 on the correct label $k = y_i$. By directly conformalizing in the probability space, we obtain an appealing calibration property for the constructed credal region: the true data generating process belongs to the credal region with high probability $1 - \alpha$, where $\alpha$ is chosen by the user. Due to this property, our credal regions can be used to disentangle and quantify aleatoric and epistemic uncertainties associated with the analysis at hand (Hüllermeier & Waegeman, 2021; Sale et al., 2023). On both synthetic and real datasets, we verify both the coverage of the constructed regions and the true label coverage guarantee of the derived predictive sets. In contrast to Cella & Martin (2022a;b), we do not require any assumption beyond exchangeability. Using imprecise highest density sets, we can derive predictive sets of labels that are more efficient, i.e., smaller on average, compared to those by Stutz et al. (2023c).

**Outline.** This paper is structured as follows: Section 2 provides the required background. First, in Section 2.1, we recall the plausibility regions and the conformal predictive sets derived in Stutz et al. (2023c) and define the corresponding conformal coverage guarantee which we built on in the following. Then, in Section 2.2 we revisit the necessary imprecise probabilistic concepts relevant for this paper, including imprecise highest density sets (IHDSs). Part of our main results, Section 3 shows that the plausibility region in Stutz et al. (2023c) is equivalent to a credal region, which also satisfies a desirable calibration property. Then, Sections 3.1 and 3.2 discuss the utility of our credal region the field of Imprecise Probabilistic Machine Learning in general and the relation and improvements over the work of Cella & Martin (2022a;b) in particular. Finally, Section 4 presents our findings on obtaining more efficient prediction sets from our credal regions compared to (Stutz et al., 2023c). We present our experimental evidence in Section 5 and conclude in Section 6.

---

[1]Reference Stutz et al. (2023b) refers to the published version, while Stutz et al. (2023c) to the first version of the paper. Since many of the concept needed in the present paper were not included in the final, published version, we differentiate between the two for ease of reference.

[2]We use lower case letters for realizations, while capital letters for random variables. In this case, calibration data $(x_1, y_1), \ldots, (x_n, y_n)$ are realizations that we observe, while the test example $X_{n+1}$ is a random variable that has not yet been realized.

[3]A pair $(x_i, \lambda_i)$ in the calibration set is the realization of a random pair $(X_i, \Lambda_i)$.

## 2 Background

### 2.1 Conformal Plausibility Regions

In Stutz et al. (2023b;c), the authors study conformal prediction for $K$-class classification in the context of *ambiguous* ground truth. Standard split conformal prediction typically assumes instead a calibration set of examples with "crisp" ground truth labels $\{(X_i, Y_i)\}_{i=1}^n$ (Shafer & Vovk, 2008). Realizing that such crisp ground truth labels might not be available in many practical settings, Stutz et al. (2023b;c) assume a calibration set of examples and so-called *plausibilities*, $\{(X_i, \Lambda_i)\}_{i=1}^n$, where plausibilities $\lambda_i$ (i.e., the realizations of $\Lambda_i$) represent categorical distributions over the $K$ possible labels.[4] This allows us to represent ambiguous examples where the corresponding distribution $\mathbb{P}[Y|X]$ is not one-hot and might have high entropy. In the full-information setting, these plausibility vectors may represent the true distribution $\mathbb{P}[Y|X]$ directly; in many practical scenarios, however, we can only approximate the true distribution. For example, disagreement among annotators frequently indicates ambiguity and deterministic or probabilistic aggregation of multiple annotators can be used as plausibilities (Stutz et al., 2023a). Then, we assume the true distribution can be obtained in the limit of infinite "faithful" annotators.

For a new test example $X_{n+1}$ and a significance level $\alpha \in [0, 1]$, a so-called *plausibility region* $c(X_{n+1})$ is derived as

$$\left( \{(x_i, \lambda_i)\}_{i=1}^n, X_{n+1}, \alpha \right) \rightsquigarrow c(X_{n+1}) \coloneqq \{\lambda \in \Delta^{K-1} : e(X_{n+1}, \Lambda_{n+1} = \lambda) \geq \tau\}, \tag{1}$$

where $\Delta^{K-1}$ is the unit simplex in $\mathbb{R}^K$, and $e(X_{n+1}, \Lambda_{n+1} = \lambda) \coloneqq \sum_{k=1}^K \lambda_k E(X_{n+1}, k)$ is a score derived from a so-called *conformity score* $E(X_{n+1}, k)$ based on the probabilistic model prediction $p_k(X_{n+1})$. Such a model (e.g. the softmax output for class $k$ of a trained neural network) approximates the posterior class probabilities, $p_k(X_{n+1}) \approx \mathbb{P}(Y_{n+1} = k \mid X_{n+1}, D)$, where $D$ denotes the training set, and it is assumed to be available. Conformity score $E$ is built so that a high score is more unlikely, hence we can interpret $E$ as a function assigning a score to the assertion "$k$ is the correct label for the realization $x_{n+1}$ of test example $X_{n+1}$", which is lower the more the pair $(x_{n+1}, k)$ "lacks conformity" with the training data. It is worth noting that alternative definitions of $e$ have also recently been explored by Javanmardi et al. (2024). The threshold $\tau$ is then chosen using a simple quantile computation on the calibration set,

$$\tau = Q\left( \{e(X_i, \Lambda_i)\}_{i=1}^n ; \lfloor \alpha(n+1) \rfloor / n \right).$$

Under the assumption that $\{(X_i, \Lambda_i)\}_{i=1}^{n+1}$ are exchangeable, following standard conformal prediction literature (Stutz et al., 2023c, Equation (13)), the plausibility region $c(X_{n+1})$ provides a coverage guarantee stating that $\mathbb{P}[\Lambda_{n+1} \in c(X_{n+1})] \geq 1 - \alpha$,[5] where $\Lambda_{n+1}$ is the unobserved true plausibility vector of the test example. If the $\lambda_i$'s in the calibration set correspond to the true distributions $\mathbb{P}[Y|X_i = x_i]$, this provides coverage with respect to the true distribution. As (Stutz et al., 2023c, Section 3.1) details, however, in many practical settings, the plausibilities are obtained from expert annotations. With finite annotators, the authors are able to give a coverage guarantee, $\mathbb{P}_{\text{agg}}[\Lambda_{n+1} \in c(X_{n+1})] \geq 1 - \alpha$, stated in terms of the distribution $\mathbb{P}_{\text{agg}}$ that explicitly captures how annotations are aggregated into plausibilities. The underlying assumption is that $\lambda_{n+1,k} = \mathbb{P}_{\text{agg}}[Y_{n+1} = k|X_{n+1}]$ where, ideally, $\mathbb{P}_{\text{agg}} \approx \mathbb{P}$, and the annotators are the same for the calibration set and the test example. In any case, annotators can "agree to disagree" such that $\lambda_{n+1}$ has high entropy in which case $x_{n+1}$ is called ambiguous. For simplicity, we ignore this caveat for the presentation of this paper and write $\mathbb{P}$ henceforth. The interested reader can find some extra details on the difference between $\mathbb{P}_{\text{agg}}$ and $\mathbb{P}$ in Appendix A.

The plausibility region $c(X_{n+1})$ is then used in (Stutz et al., 2023c, Equation (41)) to derive "plausibility-reduced" predictive sets (PRPS) for some user-chosen $\delta \in [0, 1]$,

$$\Psi(c(X_{n+1})) \coloneqq \left\{ y \in \mathcal{Y} : \exists \lambda \in c(X_{n+1}), l \in \mathcal{Y} \text{ s.t. } \sum_{i=1}^l \lambda_{\sigma_i} \geq 1 - \delta \text{ and } \exists i \leq l \text{ s.t. } \sigma_i = y \right\}. \tag{2}$$

---

[4]They may be, for example, realizations from random variables $\Lambda_i$ distributed according to a Dirichlet distribution.
[5]This probability is also on the calibration set, since the latter is used to derive $c(X_{n+1})$.

Here $\lambda^\sigma = (\lambda_{\sigma_1}, \ldots, \lambda_{\sigma_K})^\top$ corresponds to vector $\lambda$ sorted in descending order. Equation 2 tells us that $\Psi(c(X_{n+1}))$ is the set of all labels $y$ that are assigned a probability non-smaller than $1 - \delta$ (of being the correct one for the realization of $X_{n+1}$), by at least one probability vector $\lambda$ that belongs to the plausibility region $c(X_{n+1})$.

In Stutz et al. (2023c), this is also contrasted with regular conformal predictive sets (CPS) that are obtained by calibrating a threshold $\kappa$ on the per-label scores $E$ and constructing the CPS $C(X_{n+1})$ as $\{k \in \mathcal{Y} : E(X_{n+1}, k) \geq \kappa\}$. There, the latter is further adapted to allow for ambiguous examples with plausibilities $\lambda_i$ available for calibration, see (Stutz et al., 2023c, Algorithm 1). These adapted CPS $C(X_{n+1})$ are shown to be generally more efficient, i.e., smaller, compared to $\Psi(c(X_{n+1}))$, but are unable to capture or even disentangle different sources of uncertainty. We make a step towards addressing this gap by deriving narrower predictive sets that allow for uncertainty quantification and disaggregation, using methods from imprecise probability.

## 2.2 Imprecise Probability

Following Cella & Martin (2022a;b) and additional previous work on IPML (Augustin et al., 2014; Caprio & Seidenfeld, 2023; Coolen, 1992), we briefly introduce the notions of lower and upper probabilities, and also the concept of Imprecise Highest Density Sets (IHDS). These will play a pivotal role in the paper – and especially in Section 4 – as a way to derive predictive sets of labels from our credal regions, that are shown to be more efficient than the predictive sets $\Psi(c(X_{n+1}))$.

We begin by recalling that a *credal region* $\mathcal{P}$ is a convex and closed family of probabilities. Its lower envelope $\underline{P} = \inf_{P \in \mathcal{P}} P$ is called *lower probability*, while its upper envelope $\overline{P} = \sup_{P \in \mathcal{P}} P$ is called *upper probability*. Considering the assertion that the true label $Y_{n+1}$ of test example $X_{n+1}$ is included in a set $A \subseteq \mathcal{Y}$, and using $P(A) \equiv P(Y_{n+1} \in A)$, we see that the upper probability is conjugate to the lower probability, i.e. for all $A \subseteq \mathcal{Y}$, $\overline{P}(A) = 1 - \underline{P}(A^c)$, where $A^c = \mathcal{Y} \setminus A$. Hence, studying one is sufficient to then retrieve the other. Moreover, in the present paper $\mathcal{Y}$ is finite, meaning

$$\underline{P}(A) = \inf_{P \in \mathcal{P}} P(A) = \inf_{P \in \mathcal{P}} \left[ \sum_{k \in A} P(\{k\}) \right].$$

As we can see, $\underline{P}(A)$ can be calculated in polynomial time. This is important, since using the lower probability we can derive predictive sets of labels.

**Definition 1** (Imprecise Highest Density Set, Coolen (1992)). *Let $\delta \in [0, 1]$ be any significance level. Then, the $(1 - \delta)$-Imprecise Highest Density Set (IHDS) $IS_{\mathcal{P},\delta}$ associated with $\mathcal{P}$ is the subset of $\mathcal{Y}$ that satisfies the following two conditions,*

*(i) $\underline{P}(IS_{\mathcal{P},\delta}) \geq 1 - \delta$,*

*(ii) $|IS_{\mathcal{P},\delta}|$ is a minimum of all sets for which (i) holds.*

In order to compute IHDSs in practice, we need another important result – proved in (Augustin et al., 2014, Section 4.4) and de Campos et al. (1994).

**Proposition 2** (Computing $\underline{P}(A)$). *If $\underline{P}$ avoids sure loss, i.e. if $\sum_{k \in \mathcal{Y}} \underline{P}(\{k\}) \leq 1 \leq \sum_{k \in \mathcal{Y}} \overline{P}(\{k\})$, then*

$$\underline{P}(A) = \max \left\{ \sum_{k \in A} \underline{P}(\{k\}), 1 - \sum_{k \in A^c} \overline{P}(\{k\}) \right\} \tag{3}$$
$$\geq \sum_{k \in A} \underline{P}(\{k\}), \quad \forall A \subseteq \mathcal{Y}.$$

Since $\underline{P}$ is the lower envelope of a credal region, the sure loss avoidance condition is always met, as proven in Walley (1991). We make this explicit in the following Lemma.

**Lemma 3** (Lower Envelopes Avoid Sure Loss). *The lower probability $\underline{P}$ associated with $\mathcal{P}$ avoids sure loss.*

Definition 1 can be used to derive a parallel between (generic) Conformal Prediction Sets (CPSs) and IHDSs. Condition (i) tells us that $Y_{n+1}$ belongs to $\text{IS}_{\mathcal{P},\delta}$ with $P$-probability of at least $1 - \delta$, for all distributions $P$ in the credal region $\mathcal{P}$. Condition (ii) tells us that $\text{IS}_{\mathcal{P},\delta}$ is "efficient". That is, it is the narrowest subset of the label set $\mathcal{Y}$ that is able to ensure the probabilistic guarantee of condition (i). Compared to CPSs, this is a weaker guarantee: First and foremost, the conformal guarantee is uniform, holding for all possible exchangeable distributions $P$ on $\mathcal{Y}$. Second, without assumptions on the credal region $\mathcal{P}$, condition (i) ignores whether the true distribution belongs to the credal region $\mathcal{P}$.

## 3   Conformal Plausibility Regions as Credal Regions

A key contribution of this paper is relating the plausibility regions $c(X_{n+1})$ in equation 1 to the imprecise probabilistic notion of credal regions. As we will show, this leads to a remarkable synergy that allows us to construct a credal regions for $X_{n+1}$ in a conformal way, and subsequently to use IHDSs to construct predictive sets of labels. Compared to Cella & Martin (2022a;b), our credal regions provide coverage, only requiring the exchangeability of $\{(X_i, \Lambda_i)\}_{i=1}^{n+1}$. In addition, we improve over Stutz et al. (2023c): the predictive sets that we derive are more efficient, i.e., always non-broader and sometimes strictly narrower. For a start, we show that the plausibility regions $c(X_{n+1})$ are convex and closed, and thus proper credal regions.

**Proposition 4** (Properties of the Plausibility Region). *The plausibility region $c(X_{n+1})$ derived in equation 1 is convex and closed.*[6]

*Proof.* We first show that $c(X_{n+1})$ is convex. Pick $\lambda^{(1)}, \lambda^{(2)} \in c(X_{n+1})$, $\lambda^{(1)} \neq \lambda^{(2)}$, and $\beta \in [0, 1]$. Then,

$$
\begin{aligned}
e(X_{n+1}, \beta\lambda^{(1)} + (1 - \beta)\lambda^{(2)}) &= \sum_{k=1}^{K} (\beta\lambda_k^{(1)} + (1 - \beta)\lambda_k^{(2)}) E(X_{n+1}, k) \\
&= \beta \sum_{k=1}^{K} \lambda_k^{(1)} E(X_{n+1}, k) + (1 - \beta) \sum_{k=1}^{K} \lambda_k^{(2)} E(X_{n+1}, k) \\
&\geq \beta\tau + (1 - \beta)\tau = \tau.
\end{aligned}
$$

Hence, $\beta\lambda^{(1)} + (1 - \beta)\lambda^{(2)} \in c(X_{n+1})$, which proves convexity.

Let us then turn our attention to closure. We begin by showing that $e(X_{n+1}, \cdot)$ is a continuous function. Pick a converging sequence $(\lambda^{(m)}) \subseteq c(X_{n+1})$. This means that there is a probability vector $\lambda^\star$ such that $\lambda^{(m)} \to \lambda^\star$. That is, for all $\gamma > 0$, there exists $M \in \mathbb{N}$ such that $|\lambda_k^\star - \lambda_k^{(m)}| < \gamma$, for all $m \geq M$ and all $k \in \{1, \ldots, K\}$. Now, to show continuity, we prove that, for all $\epsilon > 0$, there exists $\delta_\epsilon := \epsilon / \sum_{k=1}^{K} E(X_{n+1}, k) > 0$ such that $|\lambda_k^\star - \lambda_k^{(m)}| < \delta_\epsilon$ implies that $|e(X_{n+1}, \lambda^\star) - e(X_{n+1}, \lambda^{(m)})| < \epsilon$. Indeed,

$$
\begin{aligned}
\left| e(X_{n+1}, \lambda^\star) - e(X_{n+1}, \lambda^{(m)}) \right| &= \left| \sum_{k=1}^{K} (\lambda_k^\star - \lambda_k^{(m)}) E(X_{n+1}, k) \right| \\
&\leq \sum_{k=1}^{K} \left| \lambda_k^\star - \lambda_k^{(m)} \right| E(X_{n+1}, k) \\
&< \underbrace{\frac{\epsilon}{\sum_{k=1}^{K} E(X_{n+1}, k)}}_{=: \delta_\epsilon} \sum_{k=1}^{K} E(X_{n+1}, k) = \epsilon.
\end{aligned}
$$

This proves that $e(X_{n+1}, \cdot)$ is continuous. Then, after noting that $e(X_{n+1}, \lambda) \geq \tau$, for all $\lambda \in c(X_{n+1})$ – here the weak inequality plays a crucial role in showing closure – we can conclude that $e(X_{n+1}, \lambda^\star) \geq \tau$, and so $\lambda^\star \in c(X_{n+1})$, proving that $c(X_{n+1})$ is indeed closed. □

---

[6]The topology in which $c(X_{n+1})$ is closed is specified in the proof of the statement.

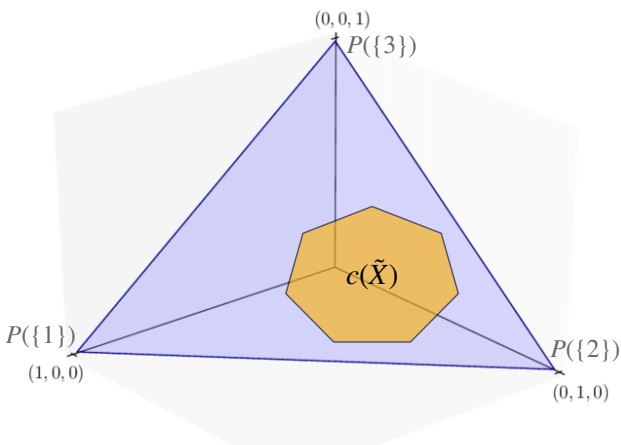

Figure 1: Suppose we are in a 3-class classification setting, so $\mathcal{Y} = \{1, 2, 3\}$. Then, any probability measure $P$ on $\mathcal{Y}$ can be seen as a probability vector. For example, suppose $P(\{1\}) = 0.6$, $P(\{2\}) = 0.3$, and $P(\{3\}) = 0.1$. We have that $P \equiv (0.6, 0.3, 0.1)^{\top}$. Since its elements are positive and sum up to 1, probability vector $P$ belongs to the unit simplex $\Delta^2$ in $\mathbb{R}^3$, the purple (2D) triangle in the figure. Then, plausibility region $c(X_{n+1})$ is a closed and convex body in $\Delta^2$, such as the depicted orange (2D) heptagon (we depict it with a solid black border to highlight the fact that it is indeed closed).

As we can see, then, every $\lambda \in c(X_{n+1})$ uniquely identifies a Categorical distribution $\text{Cat}(\lambda)$ parameterized by $\lambda$ itself. A visual representation for $c(X_{n+1})$ is given in Figure 1. A consequence of Proposition 4 is that for all $k \in \mathcal{Y}$, we can find $\underline{\lambda}_k, \overline{\lambda}_k \in [0, 1]$, $\underline{\lambda}_k \leq \overline{\lambda}_k$, such that $\lambda_k \in [\underline{\lambda}_k, \overline{\lambda}_k]$, for all $\lambda \in c(X_{n+1})$.[7] The following result relates $c(X_{n+1})$ to a credal region, that is, a closed and convex family of probabilities (Levi, 1980). It is an immediate consequence of Proposition 4 and of the definition of $c(X_{n+1})$ in equation 1.

**Corollary 4.1** (Plausibility Region as a Credal Region). *The plausibility region $c(X_{n+1})$ derived in equation 1 is equivalent to the credal region $\mathcal{P} := \{\text{Cat}(\lambda) : \lambda \in c(X_{n+1})\}$.*

Notice how $\mathcal{P}$ is a *predictive* credal region. This is because its elements can be seen as predictive distributions resulting from a Dirichlet (conjugate) prior and a Categorical likelihood. In addition, as we shall see in Section 4, $\mathcal{P}$ can be used to derive a set of labels that contain the true one for $X_{n+1}$ with high probability.

Recall that $\lambda_{n+1}$ denotes the "true" probability vector over the $K$ labels, once we observe the new test example $X_{n+1}$, and consider the Categorical distribution $\text{Cat}(\lambda_{n+1})$ parameterized by such a vector. $\text{Cat}(\lambda_{n+1})$ captures the intrinsic difficulty of labeling $X_{n+1}$. For an easy-to-categorize instance, we will have a low-entropy Categorical, and vice-versa for a highly ambiguous input $X_{n+1}$. We can also give a subjectivist interpretation to the credal region $\mathcal{P}$: we can think of $\lambda_{n+1}$ as the vector subsuming the opinions of all the experts, and that cannot be "further refined" given the available information. The sharper the disagreement between the expert around the right label for $X_{n+1}$ is, the wider $\mathcal{P}$ will be. We have the following important result.

**Proposition 5** (Probabilistic Correctness). *Let $\alpha$ be the same significance level selected in equation 1. Then,*

$$\mathbb{P}[\text{Cat}(\lambda_{n+1}) \in \mathcal{P}] \geq 1 - \alpha,$$

*where $\mathbb{P}$ depends on the statistical model relating expert opinions to plausibilities as outlined in Section 2.1 and (Stutz et al., 2023c, Equation (6)).*

*Proof.* Immediate from Corollary 4.1 and (Stutz et al., 2023c, Equation (13)). $\square$

---

[7]As a consequence, a greedy algorithm to approximate the values of $\underline{\lambda}_k, \overline{\lambda}_k$, for all $k \in \mathcal{Y}$, is easy to design, e.g. based on random samples from the uniform $\text{Unif}(c(X_{n+1}))$ on the plausibility region $c(X_{n+1})$.

### 3.1 Significance for IPML

In this section, we argue that Proposition 5 is extremely important for the field of Imprecise Probabilistic Machine Learning (IPML) (Caprio et al., 2024a; Hüllermeier & Waegeman, 2021; Zaffalon, 2002). We will further show (in the next section) that making this connection to imprecise probability allows us to derive more efficient predictive sets from these credal regions, compared to $\Psi(c(X_{n+1}))$ in equation 2.

In IPML, scholars tend to adopt either of the following two approaches. One approach is the robust statistician one, also called Huberian (Huber & Ronchetti, 2009). This is more frequentist in nature, hence the true data generating process – $\text{Cat}(\lambda_{n+1})$ in our case – is a well-defined concept. Those who take this approach proceed in two different ways: either they assume that $\text{Cat}(\lambda_{n+1}) \in \mathcal{P}$ (in which case a result like Proposition 5 is very interesting, as it allows to forego such requirement), or they verify that a calibration à la Proposition 5 holds for the methodology they propose (Acharya et al., 2015; Gao et al., 2018; Mortier et al., 2023; Liu & Briol, 2024; Chau et al., 2024). In particular, in the case of transductive conformal prediction, Martin (2023) shows that the upper envelope $\overline{\Pi}$ of the credal region $\mathcal{M}(\overline{\Pi})$ induced by conformal transducer $\pi$ as we discussed in Section 1, is the minimal outer consonant approximation of the true data generating process. This means that $\overline{\Pi}$ is the narrowest upper bound for the true distribution, that also satisfies the consonance assumption. We can then conclude – although it is not explicitly shown in their work – that credal region $\mathcal{M}(\overline{\Pi})$ contains the true data generating process. This is important in Cella and Martin's framework because it is linked to the concept of type-2 validity: since the upper probability $\overline{\Pi}$ provides control on erroneous predictions uniformly (Cella & Martin, 2022b, Definition 3), so does the true probability if the credal region is well-calibrated. Of course, we cannot directly use their result in our context because we do not have access to crisp labels. In a sense, then, in Proposition 5 we prove (a probabilistic version of) this property for the ambiguous ground truth case in classification problems.

Another approach is the deFinettian or Walleyan one (de Finetti, 1974; 1975; Walley, 1991), in which it is posited that "probability does not exist", and what we call probability measure is just a way of capturing the subjective assessments of the likelihood of events to obtain, according to the agent. In light of this, the concept of "true data generating process" is absent, and so it does not make sense to look for probabilistic guarantees à la Proposition 5.

### 3.2 Relation to Cella & Martin (2022a;b)

Given calibration data with crisp labels in $\mathcal{Y}$ as outlined in the beginning of Section 2.1, Cella & Martin (2022a;b) consider the conformal transducer $\pi : \mathcal{Y} \to [0,1]$, which assigns a $p$-value to each possible label in $\mathcal{Y}$ (Cella & Martin, 2022a, Definition 2). It uniquely identifies a credal region, assuming *consonance*. Here, $\pi$ is typically interpreted as telling us how "in line" the pair $(x_{n+1}, \tilde{y})$ is with the previously observed data $(\mathbf{x^n}, \mathbf{y^n})$. A value closer to 1 means that seeing $(X_{n+1}, Y_{n+1}) = (x_{n+1}, \tilde{y})$ would align with the preceding observations, and vice versa for values closer to 0. Once $\pi$ is obtained, consonance posits that there exists at least one label $\tilde{y}$ such that $\pi(\tilde{y}) = 1$. Loosely, this means that the label $\tilde{y}$ that makes pair $(x_{n+1}, \tilde{y})$ "the most conformal to" the observations $(\mathbf{x^n}, \mathbf{y^n})$, is required to have the highest possible value 1. This can be obtained artificially by setting $\pi$ for $\tilde{y} \in \arg\max_y \pi(y)$ to 1 (Cella & Martin, 2022a, Section 7). Then, an upper probability (that is, the upper envelope of a credal region) $\overline{\Pi}$ is derived for $\pi$ by letting $\overline{\Pi}(A) = \sup_{y \in A} \pi(y)$, for all subsets $A$ of the label space $\mathcal{Y}$. In turn, a credal region is defined as $\mathcal{M}(\overline{\Pi}) = \{P : P(A) \leq \overline{\Pi}(A)\}$, for all $A \subseteq \mathcal{Y}$. That is, $\mathcal{M}(\overline{\Pi})$ contains all the probabilities on $\mathcal{Y}$ that are set-wise dominated by $\overline{\Pi}$ where $P(A) \equiv P(Y_{n+1} \in A)$.

In comparison to Cella & Martin (2022a;b), we avoid the consonance assumption. In Cella & Martin (2022a;b), this assumptions is primarily required to bridge the so-called possibilistic approach to imprecise probabilities (Augustin et al., 2014, Chapter 4) with conformal prediction. Moreover, we do not explicitly construct the conformal transducer, also due to the fact that we work with probability vectors instead of "precise" (that is, one-hot) labels. In this sense, our work subsumes and extends Cella & Martin (2022a;b). Furthermore, our credal region $\mathcal{P}$ establishes a coverage guarantee on the true data generating process being in $\mathcal{P}$, and it also enjoys (a version of) *type-2 validity*, a notion important for uncertainty quantification, introduced for the first time in (Cella & Martin, 2022b, Definition 2). Let us be more formal about it.

**Proposition 6** (A Version of Type-2 Validity for Our Credal Region). *The credal region $\mathcal{P}$ in Corollary 4.1 is $[\delta/(1-\alpha)]$-type-2 valid, where $\alpha$ is the quantity chosen in equation 1. That is,*

$$\mathbb{P}[\overline{P}(A) \leq \delta, \, Y_{n+1} \in A] \leq \frac{\delta}{1-\alpha}, \tag{4}$$

*for all $\delta \in [0,1]$, all $n \in \mathbb{N}$, and all $A \subseteq \mathcal{Y}$. Here as in Proposition 5, $\mathbb{P}$ depends on the statistical model relating expert opinions to plausibilities.*

*Proof.* Pick any $\delta \in [0,1]$, any $n \in \mathbb{N}$, and any $A \subseteq \mathcal{Y}$. Notice that $\mathbb{P}[\overline{P}(A) \leq \delta, \, Y_{n+1} \in A] \leq \mathbb{P}[Y_{n+1} \in A]$. Now, if $\text{Cat}(\lambda_{n+1}) \in \mathcal{P}$, then $\mathbb{P}[Y_{n+1} \in A] \leq \overline{P}(A)$. In addition, given our assumption on $\delta$, we know that $\overline{P}(A) \leq \delta$. Hence, we can conclude that if $\text{Cat}(\lambda_{n+1}) \in \mathcal{P}$, then $\mathbb{P}[\overline{P}(A) \leq \delta, \, Y_{n+1} \in A] \leq \delta$. But $\text{Cat}(\lambda_{n+1}) \in \mathcal{P}$ happens with $\mathbb{P}$-probability $(1-\alpha)$ as shown in Proposition 5, and so we have that $\mathbb{P}[\overline{P}(A) \leq \delta, \, Y_{n+1} \in A] \leq \delta/[1-\alpha]$. $\square$

In words, this means that the (true/aggregated) probability of a set $A$ having small upper probability, and yet containing the true output $Y_{n+1}$ for a given new input $X_{n+1}$, is controllably small, and it depends on the parameters $\alpha$ and $\delta$ chosen by the user. When giving the bound, we need to divide by $1-\alpha$ because we need to take into account what is the $\mathbb{P}$-probability that the true distribution $\text{Cat}(\lambda_{n+1})$ belongs to the credal region $\mathcal{P}$. Notice that in equation 4 we find a slightly looser bound than the one we would derive if we were not dealing with ambiguous ground truth, i.e. if we knew that $\mathbb{P}[\text{Cat}(\lambda_{n+1}) \in \mathcal{P}] = 1$. To see that this is indeed the case, notice that if we pick $\delta = \alpha = 0.05$, then the bound in equation 4 is $\approx 0.526$, slightly larger than the one we would have (0.5) if we were not facing ambiguity.

## 4 Improving over Conformal Prediction Sets

In this section, we show how prediction set $\Psi(c(X_{n+1}))$ can be improved by using the IHDS of the lower probability $\underline{P}$ of credal region $\mathcal{P}$. In addition, we show how $\mathcal{P}$ can be used to quantify and disentangle aleatoric and epistemic uncertainties. Specifically, let us denote by $\underline{\lambda}$ the vector whose $k$-th entry is $\underline{\lambda}_k = \underline{P}(\{k\})$, and by $\overline{\lambda}$ the vector whose $k$-th entry is $\overline{\lambda}_k = \overline{P}(\{k\})$. Then, we can rewrite equation 3 as

$$\underline{P}(A) = \max \left\{ \sum_{k \in A} \underline{\lambda}_k, 1 - \sum_{k \in A^c} \overline{\lambda}_k \right\}, \quad \forall A \subseteq \mathcal{Y}. \tag{5}$$

This gives us an easy way to compute $\underline{P}(A)$ in practice, to then derive IDHSs. We can use equation 5 and Definition 1.(i) to derive Algorithm 1, a greedy algorithm to build $\text{IS}_{\mathcal{P},\delta}$. In turn, IHDS $\text{IS}_{\mathcal{P},\delta}$ is used to derive a predictive set narrower than the conformal one $\Psi(c(X_{n+1}))$ from equation 2, and that retains the same probabilistic guarantees. Recall that Definition 1.(i) gives us a probabilistic guarantee that holds for all distributions in the credal region $\mathcal{P}$ that we built in Corollary 4.1. This is a slightly weaker guarantee than that of classical conformal prediction, whose guarantee instead holds for all possible exchangeable distributions $P$ on $\mathcal{Y}$. Thanks to Proposition 5, though, a consequence of the following Proposition 7 is that the loss of coverage for the IHDS with respect to a classical CPS is negligible.

**Proposition 7** (Improving on Conformal via IPs). *Let $\alpha$ be the same significance level selected in equation 1. Pick any $\delta \in [0,1]$. Then, $\text{IS}_{\mathcal{P},\delta} \subseteq \Psi(c(X_{n+1}))$, and the inclusion is strict for some value of $\delta$. In addition, $\mathbb{P}[Y_{n+1} \in \text{IS}_{\mathcal{P},\delta}] \geq (1-\delta)(1-\alpha)$ where $Y_{n+1}$ denotes the correct label for input $X_{n+1}$.*

*Proof.* First, let us introduce the concept of $(1-\delta)$-(Precise) Highest Density Set $\text{HDS}_{P,\delta}$, for some $P \in \mathcal{P}$ Hyndman (1996),

$$\text{HDS}_{P,\delta} \coloneqq \{y \in \mathcal{Y} : P(\{y\}) \geq P^\delta\},$$

where $P^\delta$ is the largest constant such that

$$P[Y \in \text{HDS}_{P,\delta}] \geq 1-\delta.$$

By (Coolen, 1992, Page 3), we know that

$$\text{IS}_{\mathcal{P},\delta} \subseteq \bigcup_{P \in \mathcal{P}} \text{HDS}_{P,\delta}$$
$$= \{y \in \mathcal{Y} : \exists P \in \mathcal{P}, P(\{y\}) \geq P^\delta\},$$

for all $\delta \in [0,1]$, and the inclusion is strict for some value of $\delta$. The first part of the proof is concluded by noting that $\cup_{P \in \mathcal{P}} \text{HDS}_{P,\delta} = \Psi(c(X_{n+1}))$. The probabilistic guarantee according to $\mathbb{P}$ is a consequence of Proposition 5 and the fact that $\underline{P}(\text{IS}_{\mathcal{P},\delta}) \geq 1 - \delta \implies P(\text{IS}_{\mathcal{P},\delta}) \geq 1 - \delta$, for all $P \in \mathcal{P}$, by the definition of lower probability. □

In standard conformal prediction – that is, when we calibrate on crisp-labeled data – the probabilistic guarantee that we derive is of the form $\mathbb{P}[Y_{n+1} \in \text{CPS}] \geq 1 - \delta$, where we denote by CPS the conformal prediction set, and by $\delta$ the same threshold as in Proposition 7. In the present work, we need to take into account the imprecision coming from the ambiguous labeling, which appears in the form of $1 - \alpha$ in the probabilistic guarantee of Proposition 7. In our imprecise probabilistic framework, $1 - \alpha$ is the guarantee that we have on the true distribution $\text{Cat}(\lambda_{n+1})$ belonging to the credal region $\mathcal{P}$ (see Proposition 5). But $1 - \alpha$ is chosen by the user, so letting e.g. $1 - \alpha = 0.95$ or $1 - \alpha = 0.99$ will yield a negligible coverage loss with respect to the classical conformal prediction setting. It is also worth noting that the guarantee in Proposition 7 is the same derived for "standard" conformal prediction in the case of ambiguous labels in (Stutz et al., 2023c, Equation (43)).

---

**Algorithm 1** Computing Imprecise Highest Density Set $\text{IS}_{\mathcal{P},\delta}$

---

     **Input:** Vector $\underline{\lambda}$     **Parameter:** Significance level $\delta \in [0,1]$
     **Output:** IHDS $\text{IS}_{\mathcal{P},\delta}$
**Step 1** For all $A \subseteq \mathcal{Y}$, compute $\underline{P}(A)$ using equation 5.         ▷ $2^K$ possible $A$'s for $|\mathcal{Y}| = K$ labels.
**Step 2** Sort the $A$'s in ascending order of their lower probability $\underline{P}(A)$; if two or more sets have the same lower probability, put the one with lower cardinality first. Denote the sorted order as $\{A_{\rho_1}, \ldots, A_{\rho_{2^K}}\}$.
**Step 3**
**for** $k \in \{1, \ldots, 2^K\}$ **do**
   **if** $\underline{P}(A_{\rho_k}) \geq 1 - \delta$ **then**
      $\text{IS}_{\mathcal{P},\delta} = A_{\rho_k}$
      **break**
   **end if**
**end for**
**return** $\text{IS}_{\mathcal{P},\delta}$

---

Finally, let us point out that we can use credal regions to quantify and disentangle between aleatoric and epistemic uncertainties (AU and EU, respectively) in the analysis at hand (Caprio et al., 2024b). AU refers to the uncertainty that is inherent to the data generating process; as such, it is irreducible. EU, instead, refers to the lack of knowledge about the data generating process; as such, it is reducible. It can be lessened on the basis of additional data, e.g. by retraining the model using an augmented training set (Lin et al., 2024). On the other hand, since AU is irreducible, there is an increasing need for ML techniques that are able to detect and flag excess of AU, so that the user can "proceed with caution".

Recall that, for a single Categorical distribution $P = \text{Cat}(\lambda)$ on $\mathcal{Y}$, the (Shannon) entropy is defined as

$$H(P) = -\sum_{k=1}^{K} \lambda_k \log_2(\lambda_k). \tag{6}$$

The credal versions of the Shannon entropy as proposed by Abellán et al. (2006); Hüllermeier & Waegeman (2021) are $\overline{H}(\mathcal{P}) := \sup_{P \in \mathcal{P}} H(P)$ and $\underline{H}(\mathcal{P}) := \inf_{P \in \mathcal{P}} H(P)$, called the *upper* and *lower (Shannon) entropy*, respectively. The upper entropy is a measure of total uncertainty (TU), since it represents the minimum level of predictability associated with the elements of $\mathcal{P}$. In Abellán et al. (2006); Hüllermeier & Waegeman

(2021), the authors *postulate* that TU can be decomposed additively as a sum of aleatoric and epistemic uncertainties, and that the latter can be specified as the difference between upper and lower entropy, thus obtaining

$$\underbrace{\overline{H}(\mathcal{P})}_{\text{total uncertainty TU}(\mathcal{P})} \quad = \quad \underbrace{\underline{H}(\mathcal{P})}_{\text{aleatoric uncertainty AU}(\mathcal{P})} \quad + \quad \underbrace{\left[\overline{H}(\mathcal{P}) - \underline{H}(\mathcal{P})\right]}_{\text{epistemic uncertainty EU}(\mathcal{P})}. \tag{7}$$

Other measures based on credal regions are also available (see Bronevich & Rozenberg (2021); Hofman et al. (2024); Hüllermeier & Waegeman (2021) or Appendix B for a few examples) and they can be used in place of upper and lower entropy to quantify EU and AU within our credal region $\mathcal{P}$, as long as the measure chosen for the total uncertainty is bounded. We also note in passing that the decomposition in equation 7 is extremely important for the field of conformal prediction, since, as pointed out in (Hüllermeier & Waegeman, 2021, Section 5), the role of aleatoric and epistemic uncertainties in (classical) conformal prediction is in general not immediately clear. Our approach allows to overcome this shortcoming.

That being said, there is an active ongoing debate around whether TU actually decomposes *additively* into AU and EU (Baan et al., 2023; Gruber et al., 2023; Kirchhof, 2024; Ulmer, 2024; Wimmer et al., 2023). Furthermore, Mucsányi et al. (2024) convincingly proved that uncertainty measures are very context- and task-specific, and so – even under the assumption of an additive decomposition – the choice of upper and lower entropy in equation 7 is subject to the specific analysis that the user is carrying out. Because of this, we defer to future work looking for the "correct" measures for AU and EU, and how such a choice depends on the problem at hand.

Let us add a remark. In many modern-day ML and AI methodologies that allow to disentangle and quantify different types of uncertainties, the uncertainty quantification (UQ) part is not an intrinsic feature of the model, but rather something that is put "on top of" the main procedure. In contrast, uncertainty is inherent to the method we propose, via the plausibility region $c(X_{n+1})$ (and so the credal region $\mathcal{P}$). We are then able to quantify the amount of AU, and discern its types. They may even be used to build an abstaining option: if TU is "too high", the IHDS should not be returned, and instead the excess of which between AU and EU is responsible for the ambiguity should be reported.

It is worth noting that our method is not the only one exhibiting intrinsic UQ capabilities. Another notable class of such models are evidential deep learning ones (see e.g. Gao et al. (2024) for a comprehensive survey), in which uncertainty is modeled via second-order distributions, that is, using distributions over distributions. Second-order methods, though, have been recently shown to suffer from major pitfalls when used to quantify predictive EU due to their sensitivity to regularization parameters, and to underestimate predictive AU (Bengs et al., 2022; Pandey & Yu, 2023; Jürgens et al., 2024). In addition, they are not immediately comparable to credal methods because the imprecise probabilistic apparatuses that underpin them are inherently different. Indeed, contrary to second-order methods, credal regions are (convex and closed) collections of first-order probabilities.

Let us also add another comment. It is true that the results in Cella & Martin (2022a;b) could in principle be related to second-order methods. This because – through the consonance assumption $\max_{y \in \mathcal{Y}} \pi(y) = 1$ – they effectively make the upper probability $\overline{\Pi}$ of the credal region $\mathcal{M}(\overline{\Pi})$ associated to conformal prediction a consonant plausibility function. In turn, the latter corresponds to a possibility function (Zadeh, 1986), which is a second-order distribution. In this paper, instead, we do not need consonance to derive our credal region, and so the connection with second-order approaches, if it exists, is more subtle. We defer looking for it to future work.

## 5 Experiments

We verify the proposed algorithm on three datasets with ambiguous ground truth, including the toy and Dermatology DDx datasets from Stutz et al. (2023b;c) ("Toy" and "Derm", respectively), and CIFAR-10H (Peterson et al., 2019b) ("Cifar10h"). For Derm, whose details are discussed in Appendix C, we use risk labels as classes, classifying cases into low, medium and high risk. For CIFAR-10H, for computational convenience, we only consider data points whose annotated ground truth label in the original CIFAR-10 dataset is within the three classes (*airplane*, *automobile*, and *bird*). The toy dataset contains 1,000 data points, Dermatology

DDx contains 1,947 data points, and CIFAR-10H contains 3,000 data points. We split each dataset into random calibration and testing sets in a 50%-50% ratio. Furthermore, we use estimated classifier confidence scores as the conformity functions. Specifically, for the Toy dataset, we train a single-layer MLP with 100 hidden neurons, achieving an accuracy of 77.5%. For CIFAR-10H, we employ a ResNet50 (He et al., 2015) model trained on the original CIFAR-10 training set, obtaining an accuracy of 93.6%. For the Derm dataset, we rely on the confidence scores collected from the testing set reported by (Stutz et al., 2023c).

Before going on, let us pause here to add a remark on CIFAR-10H. The latter was built upon the test set of the original CIFAR-10 dataset. To reflect human perceptual uncertainty, each image in CIFAR-10H was annotated by 50 people. In our pre-processing, we first filtered out data points whose annotated labels in the original CIFAR-10 dataset were not included in {airplane, automobile, bird}. Then, for the remaining data points, we extracted the annotated probabilities of the three classes, and normalized them to sum up to one. For instance, let the original annotated probabilities be $p_{\text{airplane}}$, $p_{\text{automobile}}$ and $p_{\text{bird}}$. Then, the normalized annotated probabilities for airplane is $\frac{p_{\text{airplane}}}{p_{\text{airplane}}+p_{\text{automobile}}+p_{\text{bird}}}$. In addition to computational convenience, we focused on three classes only to be able to depict our results in the unit simplex in $\mathbb{R}^3$. As a sanity check, we also computed the runtime for each testing point across different coverage levels using 5 random seeds. As can be seen from the plot in Appendix D, the average runtime for a 3-class problem is about 1.5 ms and 3.5 for a 5-class problem, which is a reasonable increase.

We consider different miscoverage levels $\epsilon$, including 0.05, 0.1, 0.15, 0.2, 0.25 and 0.3. We denote the coverage level as $1-\epsilon$ in our plots. For simplicity, we put the significance levels $\alpha$ and $\delta$ to $\frac{\epsilon}{2}$. We compare our method (denoted by "Ours (IS)" for "Imprecise Set") to Plausibility Reduced Predictive Set $\Psi(c(X_{n+1}))$ proposed in (Stutz et al., 2023c, Equation (41)) (denoted by "PRPS"). For Algorithm 1, we construct the credal region $c(X_{n+1})$ from equation 1 by discretizing the simplex and computing the convex hull. We report three measures, including empirical distribution coverage, empirical label coverage, and average inefficiency. In particular, let $\{\lambda_1, \ldots, \lambda_N\}$ be the annotated label distributions on the $N$ testing datapoints, $\{\mathcal{P}_1, \ldots, \mathcal{P}_N\}$ be the constructed credal regions, $\{\text{IHDS}_1, \ldots, \text{IHDS}_N\}$ be the constructed Imprecise Highest Density Sets. The empirical distribution coverage is defined as

$$\frac{1}{N}\sum_{n=1}^{N}\mathbb{I}[\lambda_n \in \mathcal{P}_N], \tag{8}$$

where $\mathbb{I}[\cdot]$ denotes the indicator function. The empirical label coverage is defined as

$$\frac{1}{N}\sum_{n=1}^{N}\sum_{k=1}^{K}\mathbb{I}[k \in \text{IHDS}_N] \times \lambda_N^k, \tag{9}$$

where $\lambda_N^k$ is the annotated probability for $k$-th class. The average inefficiency is defined as

$$\frac{1}{N}\sum_{n=1}^{N}|\text{IHDS}_n|, \tag{10}$$

where $|\text{IHDS}_n|$ is the cardinality of the $n$-th IHDS. To account for randomness, we run each experiment with 20 random seeds.

**Empirical distribution and label coverage.** First, we report the empirical distribution coverage levels on different datasets in Figure 2. Since our method and the conformal-based method utilize the same algorithm to compute distribution prediction sets, we only report the true distribution coverage levels using our method. As shown in the figure, the empirical distribution coverage levels are equal to $1-\alpha$, which is consistent with Proposition 5.

Secondly, we report the end-to-end empirical label coverage levels in Figure 3. As shown in the figure, the true label coverage levels of both methods are above $1-\epsilon$, which is consistent with the coverage guarantee in (Stutz et al., 2023c, Equation (43)) and Proposition 7. Both our method and the baseline served to obtain the coverage guarantee. Furthermore, the empirical coverage is higher than the expected level because, even if the true distribution is not fully contained within the specified credal region, the resulting high density set might still include the true label. This mechanism elevates the empirical coverage above $(1-\alpha)(1-\delta)$.

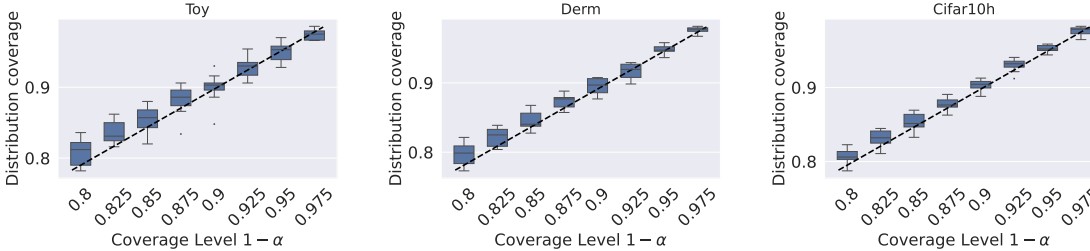

Figure 2: Empirical distribution coverage levels. The left plot is on toy dataset; the middle plot is on Dermatology DDx dataset; the right plot is on CIFAR-10H.

**Inefficiency reduction.** We report the inefficiencies using our method and the baseline in Figure 4. In our experiments, the inefficiency is defined as the average size of the resulting label prediction sets, with larger sizes indicating higher inefficiency. As shown in the plots, in most cases, our method can obtain lower inefficiencies than the baseline method. These results are consistent with Proposition 7. This implies that while both methods provide the true coverage guarantee, our method can construct smaller prediction sets that provide more information to the users. The outlier cases, where the baseline achieves lower inefficiency (e.g., $1 - \epsilon = 0.8$ on the Toy dataset), arise from discretization in the search for plausible distributions in $\mathcal{P}$, potentially introducing additional conservativeness (indeed, calibration for $1 - \epsilon$ slightly larger or lower than 0.8 removes this outlier in our experiments). Furthermore, the observed differences in inefficiency across datasets can be attributed to the variation and inherent ambiguity present within these datasets. For instance, the average inefficiency on CIFAR-10H is lower due to the concentration of probability mass on the correct label for most data points, resulting in low inherent ambiguity. In contrast, when the probability mass is more evenly distributed across labels, it induces higher inherent ambiguity.

Let us add a remark on the differences between the datasets. They stem from the underlying ambiguity of the datasets. Compared to CIFAR-10H, the Dermatology DDX dataset is significantly more challenging by having more ambiguity in the ground truth labels. This produces sets that are more conservative overall, leading to higher coverage at lower variance.

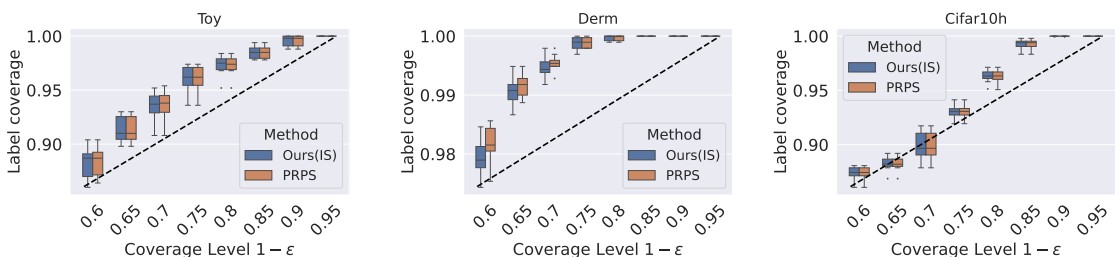

Figure 3: True label coverage levels. The left plot is on the toy dataset; the middle plot is on Dermatology DDx; the third is on CIFAR-10H.

**Effect of $\alpha$ and $\delta$.** According to Proposition 7, for a specified coverage level $1 - \epsilon$, the probability $\mathbb{P}[Y_{n+1} \in \mathrm{IS}_{\mathcal{P},\delta}]$ holds for any combination of $\alpha$ and $\delta$ as long as they satisfy $(1 - \delta)(1 - \alpha) \geq 1 - \epsilon$. However, different combinations of $\alpha$ and $\delta$ can impact the inefficiency of the resulting prediction set. To study this effect, we consider $\epsilon \in \{0.05, 0.1, 0.15, 0.2, 0.25, 0.3, 0.35, 0.4\}$, select $\alpha$ on a grid in $(0, \epsilon)$ with intervals of $\epsilon/10$, and compute the corresponding $\delta$ levels. Figure 5 shows the average inefficiency across various $\alpha$ and $\delta$ combinations. The plot indicates that $\alpha$ has a more significant effect on inefficiency; allowing higher $\alpha$ typically results in lower inefficiency (i.e., smaller prediction sets). Furthermore, given a $(1 - \alpha)(1 - \delta)$ confidence level, variations in $\alpha$ have a more pronounced impact on the resulting inefficiency. Therefore, we

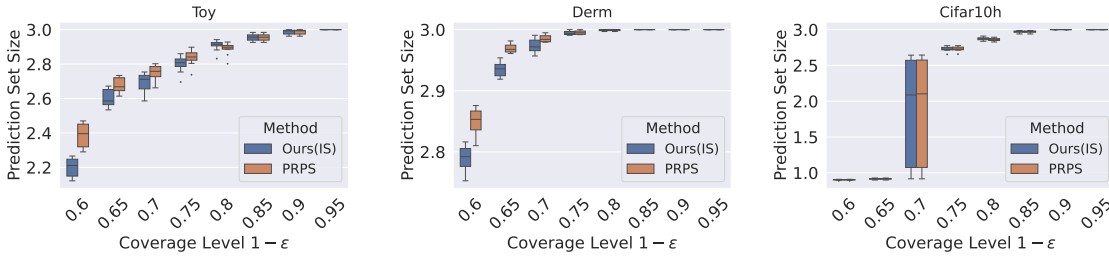

Figure 4: Average inefficiencies. The left plot is on the toy dataset; the middle plot is on Dermatology DDx; the third is on CIFAR-10H.

recommend allocating more error tolerance to $\alpha$ while maintaining a reasonable $\delta$. Developing a technique to identify the optimal combination of $\alpha$ and $\delta$ levels remains a direction for future work.

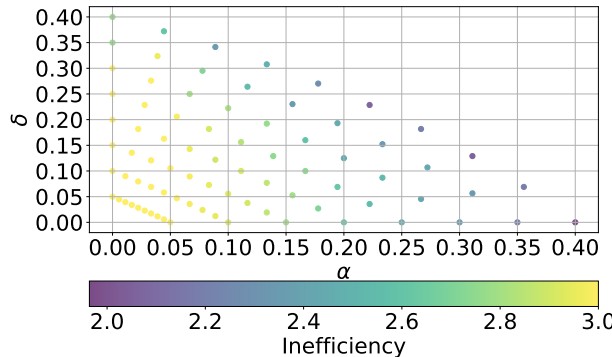

Figure 5: Inefficiency using different combinations of $\alpha$ and $\delta$.

**Qualitative analysis.** We conduct a qualitative analysis of the constructed credal regions using two examples from the DDx dataset, with results summarized in Table 1. First, as shown in the middle and right columns, the constructed credal regions (right) encompass the ground truth label distributions (middle), which were annotated by domain experts. Second, as a higher $\epsilon$ level demands greater coverage, the resulting credal regions become larger.

In addition, we also compute lower and upper entropy for both examples. For the first one, we have that $\text{AU}(\mathcal{P}) = \underline{H}(\mathcal{P}) = 0$ and $\text{TU}(\mathcal{P}) = \overline{H}(\mathcal{P}) = 1.091$, so that $\text{EU}(\mathcal{P}) = \overline{H}(\mathcal{P}) - \underline{H}(\mathcal{P}) = 1.091$. For the second one, we have that $\text{AU}(\mathcal{P}) = \underline{H}(\mathcal{P}) = 0$ and $\text{TU}(\mathcal{P}) = \overline{H}(\mathcal{P}) = 0.991$, so that $\text{EU}(\mathcal{P}) = \overline{H}(\mathcal{P}) - \underline{H}(\mathcal{P}) = 0.991$. As we can see, in both cases the entire uncertainty faced by the user is of epistemic nature. The reason why this happens is that in both cases at least one one-hot encoding probability vector belongs to the credal region $\mathcal{P}$. In particular, in both cases the credal regions contain the one-hot encoding vector for the label "Medium", $\lambda^\star = (0, 1, 0)^\top$. Then, given the current state of epistemic knowledge, we cannot exclude that if we were to collect enough extra data, we would end up finding a singleton credal region $\mathcal{P} = \{P^\star = \text{Cat}(\lambda^\star)\}$. In that case, we would have zero aleatoric uncertainty, since we would be sure (with $P^\star$-probability 1) that the label "Medium" is the correct one for the new input. In turn, since for the time being we cannot exclude this best-case scenario, all the uncertainty encoded in the credal regions is of reducible nature (EU).

# 6 Conclusion

In this paper, we address an important problem in imprecise probabilistic machine learning, namely how to empirically derive credal regions in a data-driven and efficient way, without any prior assumptions. To this end, we build on previous work by Stutz et al. (2023b;c) and apply conformal prediction in the probability

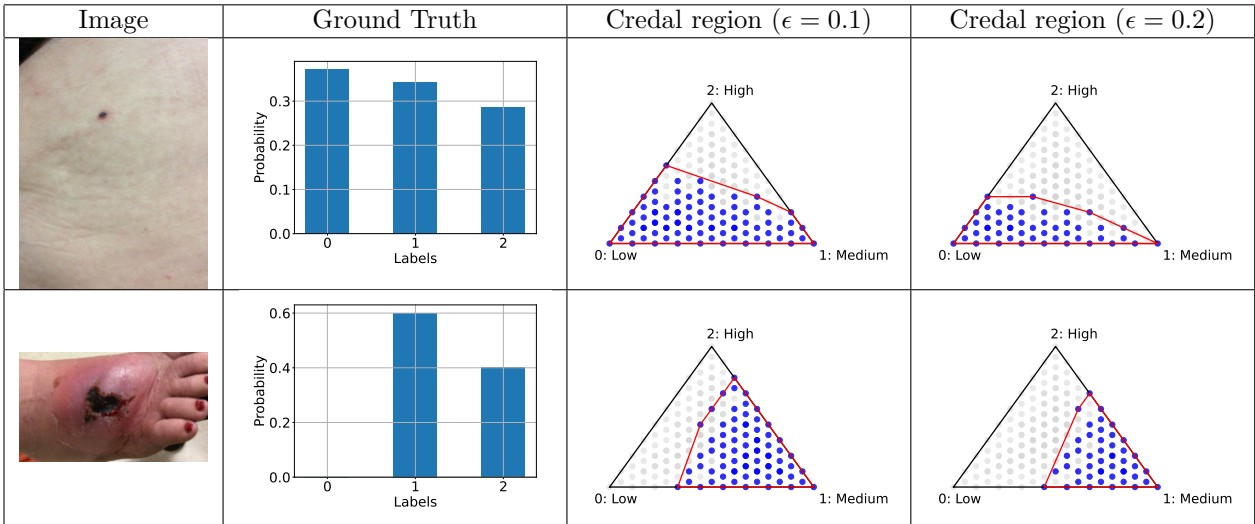

Table 1: Qualitative Examples from the DDx Dataset: The left column presents the original medical images, the middle column shows the label distributions annotated by experts, and the right column displays the credal regions constructed using our method.

space. Given a calibration set of examples with associated categorical distributions over classes, we can explicitly construct credal regions that enjoy a calibration property guaranteeing the true data generating process being included with high probability. Besides being efficient, this approach also allows to deal with ambiguous ground truth, examples where the ground truth label is uncertain. By constructing imprecise highest density sets (Coolen, 1992), we can derive predictive sets of labels that are narrower, i.e., more efficient than those from Stutz et al. (2023c). Compared to the seminal work of Cella & Martin (2022a;b), we obtain credal regions without explicitly constructing the conformal transducer (i.e., computing $p$-values for all classes of the test example) and avoid the consonance assumption (stating that the $p$-value for the class $\tilde{y}$ that makes $(x_{n+1}, \tilde{y})$ the "most conformal" to what we see in the calibration set has to be 1). Moreover, due to our calibration property, we do not need to assume that the true data generating process is included in the credal region. We also show that our credal region is $[\delta/(1-\alpha)]$-type-2 valid, a version of a notion introduced in Cella & Martin (2022b, Definition 2).

**Limitations.** Eliciting $\mathcal{P}$ requires exchangeability as a consequence of using a conformal approach; it also requires the availability of a calibration set. In addition, some subjectivity enters the analysis via the choice of the non-conformity measure for the conformal prediction methodology. Moreover, as proven e.g. in Lei & Wasserman (2013), with finite samples, we cannot give a conditional version of the probabilistic guarantee in Proposition 7. Furthermore, the discretization involved in constructing credal regions can lead to high computational complexity, particularly when the label space is large. Finally, throughout the paper we (tacitly) assumed that the plausibility vectors $\lambda_i$ in the calibration set are available and accurate; this may not be the case in some applications (Stutz et al., 2023a).

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

# A   Differences Between $\mathbb{P}_{\mathrm{agg}}$ and $\mathbb{P}$

Following the argumentation of Stutz et al. (2023b, Section 3.1), the first step is to realize that in reality, we always deal with $\mathbb{P}_{\mathrm{agg}}$ rather than $\mathbb{P}$. This is because in most cases, our labels are derived from annotations rather than from some objective process. Even if a ground truth is obtained from a measurement not involving human annotation, there might be noise and errors in these measurements. With this realization, the best we can do in terms of a coverage guarantee is to guarantee coverage with respect to $\mathbb{P}_{\mathrm{agg}}$ (since we do not know and will likely never know the true underlying $\mathbb{P}$). In other words, performing well against $\mathbb{P}_{\mathrm{agg}}$ means performing well against expert annotators. In light of $\mathbb{P}$ being unavailable, performing as good as the expert annotations is the best we can ever hope for.

Mathematically, we can model $\mathbb{P}_{\mathrm{agg}}$ using an annotation process as described in detail in the last two paragraphs of Stutz et al. (2023b, Section 3.1).

## B   Uncertainty Quantification

Other measures for AU and EU are also available in the context of credal regions (Bronevich & Rozenberg, 2021; Hofman et al., 2024; Hüllermeier & Waegeman, 2021), and they can be used in place of upper and lower entropy to quantify EU and AU within our credal region $\mathcal{P}$, as long as the measure chosen for the total uncertainty is bounded. We chose upper and lower entropy because of their ease of computation. In fact, let us give a case in which the quantities in equation 7 can be easily calculated or bounded. Let $ex\mathcal{P}$ denote the extreme elements of $\mathcal{P}$, that is, those elements that cannot be written as a convex combination of one another. We have that $\mathcal{P} = \text{Conv}(ex\mathcal{P})$, and $\text{Conv}(\cdot)$ denotes the convex hull operator. Then, the following was proven in (Caprio et al., 2024b, Theorem 8).

**Proposition 8.** *Suppose that* $|ex\mathcal{P}| = S < \infty$. *Let*

$$\underline{H}(P^{ex}) := \min_{P_s^{ex} \in ex\mathcal{P}} H(P_s^{ex})$$

*and*

$$\overline{H}(P^{ex}) := \max_{P_s^{ex} \in ex\mathcal{P}} H(P_s^{ex}).$$

*Let*

$$l[TU(\mathcal{P})] := \max\left\{ \sup_{\beta \in \Delta^{S-1}} \sum_{s=1}^{S} \beta_s H(P_s^{ex}), \overline{H}(P^{ex}) \right\},$$

*then*

$$TU(\mathcal{P}) \in \left[ l[TU(\mathcal{P})], \sup_{\beta \in \Delta^{S-1}} \sum_{s=1}^{S} \beta_s H(P_s^{ex}) + \log_2(S) \right],$$

$$AU(\mathcal{P}) = \underline{H}(P^{ex}),$$

$$EU(\mathcal{P}) \in \left[ \max\left\{ 0, l[TU(\mathcal{P})] - \underline{H}(P^{ex}) \right\}, \right.$$

$$\left. \sup_{\beta \in \Delta^{S-1}} \sum_{s=1}^{S} \beta_s H(P_s^{ex}) + \log_2(S) - \underline{H}(P^{ex}) \right].$$

Calculating $\underline{H}(P^{\text{ex}})$ and $\overline{H}(P^{\text{ex}})$ is immediate from equation 6. On the other hand, the supremum $\sup_{\beta \in \Delta^{S-1}} \sum_{s=1}^{S} \beta_s H(P_s^{\text{ex}})$ can be computed in polynomial time.[8]

## C   Dermatology Dataset Details

In Section 5, by studying the Dermatology DDX dataset, we follow (Liu et al., 2020; Stutz et al., 2023a;b;c) and consider a very ambiguous as well as safety-critical application in dermatology: skin condition classification from multiple images. Such a dataset is from Liu et al. (2020), and consists of 1949 test examples and 419 classes with up to 6 color images resized to $448 \times 448$ pixels. The classes, i.e., conditions, were annotated by various dermatologists who provide partial rankings. These rankings are aggregated deterministically to obtain the plausibilities $\lambda$ using the inverse rank normalization procedure of (Liu et al., 2020) described in (Stutz et al., 2023b, Section 3.1).

## D   Runtime of CIFAR-10H with More Classes

As a sanity check, for CIFAR-10H we computed the runtime for each testing point across different coverage levels using 5 random seeds. As can be seen from the plot below, the average runtime for a 3-class problem is about 1.5 ms and 3.5 for a 5-class problem, which we deem to be a reasonable increase.

---

[8]We also point out how bounds for $TU(\mathcal{P})$, $AU(\mathcal{P})$, and $EU(\mathcal{P})$ in terms of the *entropy of the lower probability* and of the *entropy of the upper probability*, $H(\underline{P})$ and $H(\overline{P})$, respectively – not to be confused with the lower and upper entropy $\underline{H}(\mathcal{P})$ and $\overline{H}(\mathcal{P})$, respectively – were given in (Caprio et al., 2024a, Theorem 13).

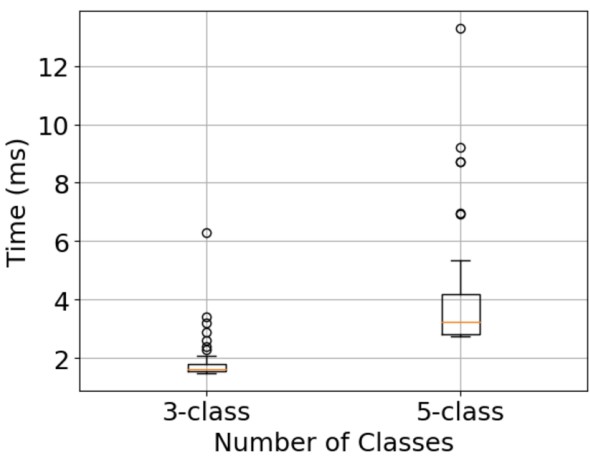

