# OpenReview forum: "Conformalized Credal Regions for Classification with Ambiguous Ground Truth"
_TMLR — Accepted by TMLR_

### Review · Reviewer_DMPv · 2024-11-17

**Summary Of Contributions:**

This paper shows that if you run conformal prediction on “plausibilities” (which can be thought of representing P(Y|X_i)), then the resulting sets of distributions are credal regions, meaning that they are convex and closed. Taking the Imprecise Highest Density Set of such a credal region can yield smaller prediction sets compared to the method of Stutz et al. 2023c (shown empirically) while not compromising coverage significantly (shown theoretically in Prop. 7 and also empirically).

**Audience:**

Yes

**Claims And Evidence:**

Yes

**Requested Changes:**

Experiments section:
* I’d like to see a more detailed description of how the plausibilities for each dataset are generated (could go in the Appendix). Were they expert annotators? How many annotators? For CIFAR-10, how did you deal with probability mass assigned to classes you did not include?
* You say that “for computational convenience” you only run the experiments using 3 classes — can your algorithm not be easily run with more than 3 classes? Can you give a sense of run time (minutes? hours?) of your current experiments? I’d like to see experiments using more classes if it is not prohibitively hard. If it is too hard, then you should acknowledge the computational burden of your method in your Limitations paragraph at the end of the paper.
* Can you write out mathematical definitions of the metrics you use? I presume that “empirical distribution coverage” refers to the fraction of times the credal region contains $\lambda_i$ and “empirical label coverage” refers to the fraction of times the IHDS derived from the credal region contains $Y_i$.
* I’d like to see the coverage plot with a dotted y=x line to make it easier to compare the desired coverage level to the realized coverage level.
* Do you have explanations for the cause of differences between the different datasets? For example, why does the Derm dataset have such high label coverage and such small variance in set sizes as we vary the desired coverage level?

Small changes:
* I don’t believe “this allows to represent ” is grammatically correct (bottom of p. 2). Perhaps “this allows us to represent” or “this allows the representation of.” The same applies to the first paragraph of Sec 4.1
* Eq (2) - has $\delta$ been defined by this point? You can just add “for some user-chosen $\delta \in [0,1]$."
* Eq (2) requires quite a bit of effort to parse, so it would be good to describe it in words or provide an example of how a PRPS is constructed given a plausibility region.
* In the Experiments section, why is PRPS called “[49] PRPS”?

Some light suggestions. (Feel free to follow them if you think they make sense, but they are not required changes.)
* As mentioned above, the discussion of using the credal regions to identify aleatoric vs. epistemic uncertainty was great. If there are any compelling case studies/examples that can be extracted from the experiments you ran, that would be interesting to see. Perhaps this can be applied with the qualitative analysis in Table 1.
* You could consider creating combining Sec 2 and Sec 3 into one section called “Background.” That would make it a bit easier to distinguish between existing work and your contributions.
* I would appreciate some more discussion of how to interpret the probabilistic guarantees if $P_{\mathrm{agg}}$ is significantly different from the true data-generating $P$ and whether this knowledge should affect how we choose to “split” $\epsilon$ between $\delta$ and $\alpha$.
* Are there any non-conformal methods for generating credal regions that are relevant to compare against in your experiments?

**Strengths And Weaknesses:**

Strengths:
* Overall, the paper is well-written.
* Good description of relevant background on Imprecise Probability.
* The discussion of how conformalized credal regions can be used to disentangle aleatoric and epistemic uncertainty is very interesting.
* I appreciate the specificity when referring to related work (the section and equation numbers are helpful).

Weaknesses: The experiments section could be improved. See Requested Changes.

---

> ### Author Response · Authors · 2024-12-31
>
> First, we thanked the reviewer for providing this constructive feedback. We address the questions and concerns below:
>
> ## 1, More details on how each dataset is generated
> We added the dataset details on CIFAR-10H dataset after the first paragraph of Section 5; and details on the Dermatology dataset in Appendix C. Please refer to [1] for details on the toy dataset.
>
> ## 2, More classes
> We computed the runtime for each testing point across different coverage levels using 5 random seeds.
> The results are shown in Appendix D. As can be seen from the plot, the average runtime for a 3-class problem is about 1.5 ms and 3.5 for a 5-class problem, which is reasonable.
>
> ## 3, Mathematically definitions of metrics
> We added the mathematical definitions of our metrics in third paragraph of Section 5.
>
> ## 4, Add dotted y=x
> We thank the reviewer for proposing this change. We have added the suggested line in the updated plots.
>
> ## 5, Explanations for differences across datasets
> We added the explanation for the differences at the end of the last paragraph of page 11.
>
> ## 6, Problem with [49] PRPS
> We refer to [49] since these sets were previously considered in Stutz et al., 2023c which previously was reference [49]. We forgot to update the plot after a template change. We fixed this in the updated version.
>
> ## 7, Examples on identifying aleatoric vs epistemic
> We added the example and discussed the difference between aleatoric and epistemic uncertainties in the last paragraph of page 12.
>
> ## 8, Combining Sec 2 and Sec 3
> We thank the reviewer for suggesting this change, which will be implemented in the final version of our paper.
>
> ## 9, Discussions on $\mathbb{P}_{agg}$
> This is a key question in light of working with annotators. Following the argumentation of Stutz et al., 2023, Section 3.1, the first step is to realize that in reality, we always deal with $\mathbb{P}\_{agg}$ rather than $\mathbb{P}$. This is because in most cases, our labels are derived from annotations rather than from some objective process. Even if a ground truth is obtained from a measurement not involving human annotation, there might be noise and errors in these measurements. With this realization, the best we can do in terms of a coverage guarantee is to guarantee coverage wrt. to $\mathbb{P}\_{agg}$ (since we don’t know and will likely never know the true underlying $\mathbb{P}$). In other words, performing well against $\mathbb{P}_{agg}$ means performing well against expert annotators. Note that in light of $\mathbb{P}$ being unavailable, performing as good as the expert annotations is the best we can ever hope for.
>
> Mathematically, we can model $\mathbb{P}_{agg}$ using an annotation process as described in detail in Stutz et al., 2023, Section 3.1 (last two paragraphs).
>
> ## 10, Non-conformal methods for generating credal regions
> We thank the reviewer for asking such a good question. The answer is yes; for example, we could compare against the (predictive) IHDS resulting from Credal Bayesian Deep Learning (https://openreview.net/forum?id=4NHF9AC5ui). The reason why we do not compare against such methods is twofold. First, typically Imprecise Probabilistic Machine Learning (IPML) methods do not provide coverage guarantees à la Propositions 5 and 6 in our paper. Second, they typically are model-based methods, in which the user has to specify e.g. (a set of) priors and (a set of) likelihoods. Instead, since the method we propose hinges on Conformal Prediction, it is model-free. For these two main reasons, we feel a direct comparison with non-conformal methods is unfair, at least given the current state of IPML research.
>
> ## 11, Other suggested minor changes
> We thank the reviewer for spotting this mistake, which will be mended in the final version of our paper.
>
> [1] Stutz et al., Conformal prediction under ambiguous ground truth, 2023.
>
> Please kindly let us know if you have any further questions or concerns. We are happy to provide further clarifications.

---

> > ### Comment · Reviewer_DMPv · 2025-01-10
> > **Comment on author response**
> >
> > I thank the authors for the work they have put in to address the reviewer comments. I appreciate the additional explanations and I especially appreciate the new runtime experiments. I believe the changes have improved the clarity of the paper.

---

### Review · Reviewer_9sQe · 2024-11-18

**Summary Of Contributions:**

The authors study the task of producing prediction sets for classification tasks with K classes, and the training set has label noise as quantified by plausibility vectors that are assumed accurate and available.

A direct application of conformal prediction produces prediction sets over label distributions with coverage level $1-  \alpha$. Previous works has reduced this into a prediction set over labels by taking the union over Highest Density Sets (equation 2 and proof for Proposition 7). The authors instead use of Imprecise Highest Density Sets of level $1-\delta$, which gives potentially improved tightness, at the cost of  controllable miscoverage.

The derivation is connected tightly to the litterature of Imprecise Probability, and framed in terms of Credal Regions. The authors show that the prediction set produced by the conformal prediction step is a credal region with typ 2 validity. Being a credal set, it unlocks theory for analyzing aleatoric vs epistemic uncertanity.

**Audience:**

Yes

**Broader Impact Concerns:**

I don't think one is needed.

**Claims And Evidence:**

Yes

**Requested Changes:**

**Crititcal changes**

The phrasing in the abstract "a closed and convex family of probabilities on the output space" threw me off initially. I think many other readers will be similarly startled. More precisely would be stating "a closed and convex family of probability measures on the output space", or similar.

Rewrite the proof of proposition 6. It was hard to read due to its succintness, and I believe there is a minor mistake in it due to a factor $1-\alpha$ being dropped.

I believe that the definitions of upper and lower shannon entropy should have symbols $\bar H(\mathcal P)$ not $\bar H(P)$ due to being functions on sets of measures.

**Changes that would strengthen the work**

"Inefficiency" in the experiment seems like a misnomer. A better measure of inefficiency would be the number of excess labels in the prediciton set, compared with an oracle. This is doable for the toy data set. I recommend you to do this. Keep the prediction set size that you have now computed, but rename it to "Prediction set size" or similar.

**Strengths And Weaknesses:**

**Strong points**

The submission connects the fields of conformal prediction and imprecise probability. This is very good.

They show a way to deal with label noise in conformal prediction. This is interesting.

The conclusions section 7 honestly reports the merits of the paper. Overall the paper seems fair and well representing the contents of it.

The experimental section corroborates the claims of the paper.

**Weak points**

The presentation is a little convoluted to me, (coming from a background in conformal prediction, not Imprecise Probability). It seems like the authors introduce a lot of theory and concepts that are not really needed to show the main result of Proposition 7.

The algoroithm to compute the prediction sets are exponential in the number of labels. In numerical experiments, only 3 labels are considered. This is a very severe limitation, since the application of machine learning to healthcare (mentioned in the introduction) may well use more than hundreds of labels when predicting diagnoses. A discussion on the practical usefulness of the algoriothm would strengthen the work, although arguably falling outside the claims of the article.

The discussion about aleatoric and epistemic uncertanity is quite weak. It discusses this only in the narrow sense of using upper and lower shannon entropy to quanify these uncertanities. It is not investigated in numerical experiments. I think the paper would improve in focus and not lose much by dropping that section. Still - I think that the paper clearly delineates this discussion from the rest, and that the authors do not oversell their claims. Thus, I don't see a big problem with keeping it either.

---

> ### Author Response · Authors · 2024-12-31
>
> ## 1, Concerns on the presentation
> We understand the concerns of the reviewer, who we thank for their input. Let us try to persuade them that the concepts introduced prior to Proposition 7 are not superfluous. Plausibility region $c(X_{n+1})$ is needed because it is in a 1:1 relationship with our credal region $\mathcal{P}$. “Plausibility-reduced” predictive set $\Psi(c(X_{n+1}))$ is our baseline. The concept of lower probability $\underline{P}$ is instrumental to define Imprecise Highest Density Set $IS_{\mathcal{P},\delta}$, which is our main contribution. Proposition 2 is used in our Algorithm 1 to derive $IS_{\mathcal{P},\delta}$, and Lemma 3 is a sanity check. Proposition 4 is indeed not strictly needed (in the main body of the paper), but we left it in because – even without glancing at the proof – makes it clear to the reader that $c(X_{n+1})$ shares properties (closure and convexity) with $\mathcal{P}$. This intuition is strengthened by Corollary 4.1 that shows their 1:1 relationship. Propositions 5 and 6 are important because they set our method apart from classical split Conformal Prediction, and classical Imprecise Probabilistic Machine Learning models.
>
> ## 2, Concerns on the number of classes
> We computed the runtime for each testing point across different coverage levels using 5 random seeds.
> The results are shown in Appendix D. As can be seen from the plot, the average runtime for a 3-class problem is about 1.5 ms and 3.5 for a 5-class problem, which is reasonable.
>
> ## 3, Discussion about aleatoric and epistemic uncertainties
> We thank the reviewer for pointing this out. We left the discussion on aleatoric and epistemic uncertainties in the main body of the paper for a specific reason. Although it is often claimed that Conformal Prediction (CP) is an uncertainty *quantification* method, it is instead an uncertainty *representation* technique. This is because the uncertainty is represented via the conformal prediction region, whose size depends on the uncertainty faced by the user. That being said, CP does not *quantify* the (different types of) uncertainty faced by the user. In other words, it does not attach a real number to the uncertainty. On the contrary, our model can do so in light of the credal region that we derive, and of the abundant literature on credal uncertainty quantification. We only included upper and lower Shannon entropy for ease of exposition, but – as we mention in the discussion section – the user can choose between a plethora of alternatives (that we reference in the paper), provided that the measure chosen for the total uncertainty is bounded. In the updated version of our paper, we also computed TU, EU, and AU for the qualitative examples in Table 1.
>
> ## 4, Concerns on “a closed and convex family of probabilities on the output space”
> We thank the reviewer for spotting this mistake, which has been amended in the new version of the paper.

---

> > ### Author Response · Authors · 2024-12-31
> >
> > ## 5, Rewrite the proof of proposition 6
> > We thank the reviewer for this suggestion. We changed the proof as follows.
> >
> > Pick any $\delta \in [0,1]$, any $n\in\mathbb{N}$, and any $A \subseteq \mathcal{Y}$. Notice that $\mathbb{P}[\overline{P}(A) \leq \delta \text{, } Y_{n+1} \in A] \leq \mathbb{P}[Y_{n+1} \in A]$. Now, if $\textup{Cat}(\lambda_{n+1})\in\mathcal{P}$, then $\mathbb{P}[Y_{n+1} \in A] \leq \overline{P}(A)$. In addition, given our assumption on $\delta$, we know that $\overline{P}(A) \leq \delta$. Hence, we can conclude that if $\textup{Cat}(\lambda_{n+1})\in\mathcal{P}$, then $\mathbb{P}[\overline{P}(A) \leq \delta \text{, } Y_{n+1} \in A] \leq \delta$. But $\textup{Cat}(\lambda_{n+1})\in\mathcal{P}$ happens with $\mathbb{P}$-probability $(1-\alpha)$ as shown in Proposition 5, and so we have that $\mathbb{P}[\overline{P}(A) \leq \delta \text{, } Y_{n+1} \in A] \leq \delta/[1-\alpha]$.
> >
> > In addition, we added the following explanation.
> >
> > In words, this means that the (true/aggregated) probability of a set $A$ having small upper probability, and yet containing the true output $Y_{n+1}$ for a given new input $X_{n+1}$, is controllably small, and it depends on the parameters $\alpha$ and $\delta$ chosen by the user. When giving the bound, we need to divide by $1-\alpha$ because we need to take into account what is the $\mathbb{P}$-probability that the true distribution $\textup{Cat}(\lambda_{n+1})$ belongs to the credal set $\mathcal{P}$. Notice that in Eq.4 we find a slightly looser bound than the one we would derive if we were not dealing with ambiguous ground truth, i.e. if we knew that $\mathbb{P}[\textup{Cat}(\lambda_{n+1})\in \mathcal{P}] = 1$. To see that this is indeed the case, notice that if we pick $\delta=\alpha=0.05$, then the bound in Eq.4 is $\approx 0.526$, slightly larger than the one we would have ($0.5$) if we were not facing ambiguity.
> >
> > ## 6, Suggestions on upper and lower Shannon
> > We thank the reviewer for this suggestion. We followed the notation put forth e.g. in https://www.tandfonline.com/doi/abs/10.1080/03081070500473490, https://link.springer.com/article/10.1007/s10994-021-05946-3, and https://openreview.net/forum?id=4NHF9AC5ui. But the reviewer is right, and we amended the notation in the new version of the paper.
> >
> > ## 7, Concerns about “Inefficiency”
> > First, we thank the reviewer for suggesting changing 'inefficiency' to 'prediction set size.' We have incorporated this update in our revision. We also appreciate the suggestion of the new metric, 'excess labels.' However, we note that the ground truth labels in our problem are inherently ambiguous, rendering oracle prediction sets ill-defined. This issue of ambiguous ground truth remains present even in the toy dataset. Moreover, we emphasize that inefficiency—quantified by the size of the prediction sets—is meaningful in both unambiguous [1] and ambiguous ground truth settings [2]. Consequently, we believe reporting the prediction set size, rather than excess labels, is more appropriate for our problem."
> >
> >
> > [1] A Angelopoulos, S Bates. A Gentle Introduction to Conformal Prediction and Distribution-Free Uncertainty Quantification, 2022.
> > [2] D Stutz, et al. Conformal prediction under ambiguous ground truth, 2023.
> >
> > We are happy to provide further clarifications on any clear points or further questions.

---

### Review · Reviewer_6HLc · 2024-12-09

**Summary Of Contributions:**

The authors provide a nove method to create conformalized credal regions, i.e. convex sets of distributions on the probability simplex that contain the true (ambiguous) distribution for an input in expectation. To this end, they build on former work on Conformal plausibility regions, which combined with concepts from imprecise probabilities yields conformalized credal regions. They provide formal motivations and proofs for their idea and validate it on three datasets, including a toy and a Dermatology dataset and CIFAR-10H.

**Audience:**

Yes

**Broader Impact Concerns:**

I foresee no direct ethical implication with the reviewed work.

**Claims And Evidence:**

No

**Requested Changes:**

* I would very much like to see a more comprehensive related work section on similar works in conformal prediction and credal sets, with potential connections to evidential deep learning as another family of second-order methods, if applicable.
* Test the disentanglement of uncertainty into aleatoric and epistemic using equation 6 empirically. This could for instance be tested using misclassification or OOD detection as a proxy task. The authors claim their method is well-suited for this disentanglement (see contributions paragraph in section 1), but no empirical evidence is given to support this claim.
   * This part might also warrant a slight critique of the aleatoric / epistemic decomposition, the limits of which have been argued from a multitude of perspective, see for instance the works by [1-5]
   * I don't fully agree with the remark at the end of section 5; while I (think i) understand its intention, I think this remark is a bit too general. You could argue that methods like the aforementioned evidential deep learning also come with uncertainty "built in", and thus I would request the authors to sharpen this statement and provide more details / references about the types of methods they criticize here.
* Explain some of the empirical results in more detail.
  * As far as I can see, it was not reported which underlying models and conformity function were used in the experiments in section 6.
  * I would also like to understand better why there seems to be a gap between desired and actual coverage in the plots in Figure 3, the potential reasons for the wide distribution of inefficiency values in the third plot in Figure 4 for 0.7 desired coverage.
  * The small number of datapoints used for experiments (<= 3k) and the reduction of the number classes in CiFAR10H to three for "computational convenience" makes me suspect that the expontential number of searches in algorithm 1 is computationally prohibitive, which should be reported in the limitations section.
* Some additional information / intuition on how to choose $\alpha$ and $\delta$ in practice beyond the analysis in section 6.
* Some minor typos / issues:
    * Uppercase "equation" at the end of the first paragraph in 4.1 and another spots in the paper.
    * In the proof of proposition 4, I would suggest to replace the notation $\lambda^1$ with $\lambda^{(1)}$ to avoid confusion. Also, I believe that in the same proof the quantity $\lambda^*$ is not properly introduced or explained.
    * Some of the citations including page or section numbers feel a bit wonky, take for instance page 8 "By (Coolen, 1992, Page 3)" when it feels more natural to have "By Coolen (1992), p. 3" or similar. This occurs multiple times throughout the paper.
    * In the equation under this exact citation, I believe a "P({y})" was erroneously denoted as "P(y)".
    * The last paragraph in section 6 seems a bit to superficial to be called qualitative analysis, although I really like the examples shown.
* I am not sure if this technically violates the requirements of TMLR, but the appendix before the bibliography feels a bit unnatural to me.
* The authors cite Stutz (2023a;b;c) as work learning from ambiguous ground truth. However, ambiguous ground truth has also been studied in many works in Natural Language Processing already [6,7], with a more comprehensive account in [8], and examples in computer vision as well [9].

[1] Baan, Joris, Nico Daheim, Evgenia Ilia, Dennis Ulmer, Haau-Sing Li, Raquel Fernández, Barbara Plank, Rico Sennrich, Chrysoula Zerva, and Wilker Aziz. "Uncertainty in natural language generation: From theory to applications." arXiv preprint arXiv:2307.15703 (2023).

[2] Gruber, Cornelia, Patrick Oliver Schenk, Malte Schierholz, Frauke Kreuter, and Göran Kauermann. "Sources of Uncertainty in Machine Learning--A Statisticians' View." arXiv preprint arXiv:2305.16703 (2023).

[3] Kirchhof, Michael. "Uncertainties of latent representations in computer vision." arXiv preprint arXiv:2408.14281 (2024).

[4] Ulmer, Dennis. "On Uncertainty In Natural Language Processing." arXiv preprint arXiv:2410.03446 (2024).

[5] Wimmer, Lisa, Yusuf Sale, Paul Hofman, Bernd Bischl, and Eyke Hüllermeier. "Quantifying aleatoric and epistemic uncertainty in machine learning: Are conditional entropy and mutual information appropriate measures?." In Uncertainty in Artificial Intelligence, pp. 2282-2292. PMLR, 2023.

[6] Nie, Yixin, Xiang Zhou, and Mohit Bansal. "What can we learn from collective human opinions on natural language inference data?." arXiv preprint arXiv:2010.03532 (2020).

[7] Baan, Joris, Wilker Aziz, Barbara Plank, and Raquel Fernandez. "Stop measuring calibration when humans disagree." arXiv preprint arXiv:2210.16133 (2022).

[8] Plank, Barbara. "The'Problem'of Human Label Variation: On Ground Truth in Data, Modeling and Evaluation." arXiv preprint arXiv:2211.02570 (2022).
[9] Peterson, Joshua C., Ruairidh M. Battleday, Thomas L. Griffiths, and Olga Russakovsky. "Human uncertainty makes classification more robust." In Proceedings of the IEEE/CVF international conference on computer vision, pp. 9617-9626. 2019.

**Strengths And Weaknesses:**

Strengths
-----------

* The paper is mostly well-written, includes formal proofs and gives intution about some of the theoretical concepts, and is thus a very educational resource on the topic.
* The contribution is an important step towards credal sets with statistical guarantees.

Weaknesses
--------------

* The paper is missing a more explicit review of related work in credal sets, conformal prediction, and potentially also evidential deep learning.
* The authors reiterate a postulated decomposition into aleatoric and epistemic uncertainty amnd claim that their method is well-suited for this decomposition, but it is never tested in experiments in the paper.
* Some of the experimental results are not discussed in a lot of detail.

---

> ### Author Response · Authors · 2024-12-31
>
> ## 1, More comprehensive related work section
> We thank the reviewer for their suggestion. In the paper, we cite and relate our findings to all the existing papers that study the connection between credal sets and conformal prediction. In particular, we cite Javanmardi, Stutz, and Hüllermeier 2023 that discusses the relationship between a credal approach like ours and second-order methods like evidential deep learning. Let us also be clearer about second-order methods. These are not (directly) comparable to credal sets because the imprecise probabilistic apparatuses that underpin them are inherently different. Credal sets are (convex and closed) collections of first-order probabilities, while evidential deep learning is rooted in second-order probabilities, that is, probabilities over probabilities. We report the discussion on this matter that we added to our manuscript in the following two paragraphs.
>
> It is worth noting that our method is not the only one exhibiting intrinsic UQ capabilities. Another notable class of such models are evidential deep learning ones (see e.g. Gao et al. (2024) for a comprehensive survey), in which uncertainty is modeled via second-order distributions, that is, using distributions over distributions. Second-order methods, though, have been recently shown to suffer from major pitfalls when used to quantify predictive EU due to their sensitivity to regularization parameters, and to underestimate predictive AU (Bengs et al. 2022; Pandey & Yu, 2023; Jurgens et al., 2024). In addition, they are not immediately comparable to credal methods because the imprecise probabilistic apparatuses that underpin them are inherently different. Indeed, contrary to second-order methods, credal sets are (convex and closed) collections of first-order probabilities.
>
> Let us also add another comment. It is true that the results in Cella & Martin (2022a;b) could in principle be related to second-order methods. This because -- through the consonance assumption $\max_{y \in \mathcal{Y}} \pi(y)=1$ -- they effectively make the upper probability $\overline{\Pi}$ of the credal set $\mathcal{M}(\overline{\Pi})$ associated to conformal prediction a consonant plausibility function. In turn, the latter corresponds to a possibility function (Zadeh, 1986), which is a second-order distribution. In this paper, instead, we do not need consonance to derive our credal region, and so the connection with second-order approaches, if it exists, is more subtle. We defer looking for it to future work.

---

> > ### Author Response · Authors · 2024-12-31
> >
> > ## 2, Disentanglement of uncertainty into aleatoric and epistemic
> > We thank the reviewer for this suggestion.
> >
> > While our method has uncertainty quantification capability, that is not the main point of the paper. We simply point out that measures for epistemic and aleatoric uncertainty (EU, AU) based on credal sets exist (such as the upper and lower entropy discussed in the paper), and can be readily used to quantify EU and AU in the analysis at hand, when using our method. In a sense, we get UQ capabilities “for free” as a result of building the credal region $\mathcal{P}$. As a consequence, we do not feel the suggested experiments are in order for the present work, especially in light of the recent work https://openreview.net/forum?id=x8RgF2xQTj#discussion, that makes it clear that uncertainty measures are very context- and task-specific. Rather, they will be the specific topic of a follow-up paper, in which we look in detail into uncertainty decomposition in CP with ambiguous ground truth.
> >
> > We will change “well-suited” with “can be used to” in the new version. Let us also add two comments. First, we are well aware of the existing literature that criticizes the additive decomposition of total uncertainty into EU and AU. We will make this clearer in the new version, by also citing the suggested references. Once again, we only wanted to point out that the credal set that we derive can be “readily used” to quantify AU and EU. We have no claim that the measures we propose are the best ones, and we tried to make it clear with the statement right below equation (7). We will be more precise in the new version, and also add a citation to the above-mentioned recent paper https://openreview.net/forum?id=x8RgF2xQTj#discussion. The added paragraph in the new version of the manuscript is the following one:
> >
> > “That being said, there is an active ongoing debate around whether TU actually decomposes **additively** into AU and EU Baan et al. 2023; Gruber et al. 2023; Kirchhof, 2024; Ulmer, 2024; Wimmer et al., 2023. Furthermore, Mucsanyi et al. (2024) convincingly proved that uncertainty measures are very context- and task-specific, and so -- even under the assumption of an additive decomposition -- the choice of upper and lower entropy in Eq.7 is subject to the specific analysis that the user is carrying out. Because of this, we defer to future work looking for the ``correct'' measures for AU and EU, and how such a choice depends on the problem at hand.”
> >
> > Second, we agree with the reviewer that our remark is too general. We will cite the fact that evidential deep learning (EDL) methods too have UQ as an intrinsic feature, but we will point out – as we pointed out in our answer to the comment above – that second-order methods such as EDL ones are known to suffer from shortcomings when it comes to uncertainty quantification.

---

> > > ### Author Response · Authors · 2024-12-31
> > >
> > > ## 3, More detail on empirical results
> > > We appreciate the reviewer for highlighting these omissions. To address these concerns, we have added details on the underlying models and conformity scores to the first paragraph of the experiment section. We have also discussed why the empirical coverage levels exceed the expected values when analyzing empirical distributions and label coverage. Briefly, this can occur because the true label may appear in high-density sets even if the true distributions are not fully encompassed by the credal region.
> > > Regarding the concerns about computational complexity, we have acknowledged these limitations in our limitations section. In addition, we have included an experiment with a larger number of class labels in Appendix D, which shows that the computation time remains reasonable when scaling our method to five classes.
> > >
> > > ## 4, Additional information/intuition on how to choose $\alpha$ and $\delta$
> > > As shown in Figure 5, given a (1 - \alpha)(1 - \delta) confidence level, variations in \alpha have a more pronounced impact on the resulting inefficiency. Therefore, we recommend allocating more error tolerance to \alpha while maintaining a reasonable \delta. Developing a technique to identify the optimal combination of \alpha and \beta levels remains a direction for future work. We have added this discussion to the analysis section on the effect of \alpha and \delta.
> > >
> > > ## 5, Minor typos/issues
> > > We thank the reviewer for their suggestions. Lower-cased “equation” is due to the command \eqref in the TMLR style, so we cannot use a capital letter E. In the proof of Proposition 4, $\lambda^\star$ is introduced because we need to consider a sequence that converges, and we denote the value it converges to by $\lambda^\star$. We will make this clearer in the new version. We will implement all the other suggestions.
> > >
> > > ## 6, Concerns on the bibliography
> > > We will change this in the new version.
> > >
> > > ## 7, Additional references
> > > We will add the references mentioned by the reviewer in the new version.
> > >
> > > Let kindly let us know if you have any further questions or concerns. We are happy to make further clarifications.

---

> > > > ### Comment · Reviewer_6HLc · 2025-01-09
> > > > **Comment on response to other points of feedback**
> > > >
> > > > I thank the authors for including more information about empirical results and their method. I believe this will make the paper stronger and more informative to the community.

---

> > ### Comment · Reviewer_6HLc · 2025-01-09
> > **Comment on improved related work**
> >
> > Thank you for these additions! Given the qualitative difference of EDL to credal sets, this seems sufficient.

---

> ### Comment · Reviewer_6HLc · 2025-01-09
> **Comment on response about uncertainty disentanglement**
>
> I appreciate the authors' effort to rephrase some relevant parts of their work regarding the uncertainty disentanglement. Regarding their added paragraph, I would like to draw attention to the fact that my suggested works don't just criticize the *additive* decomposition of uncertainty into AU and EU, but also the dichotomous categorization of uncertainty into *only* AU and EU. Instead, uncertainties have more blurry boundaries and might be rather put on a spectrum from aleatoric to epistemic.

---

> > ### Author Response · Authors · 2025-01-10
> > **Thank you!**
> >
> > We thank the reviewer for their further input, with which we completely agree. So much so that we tried to exactly convey this point of view in the added paragraph. Indeed, we specifically say that uncertainty quantification is task- and context-specific, hence what is aleatoric in some cases may be epistemic in others, and vice versa. This was first pointed out in 2021 by Hullermeier and Waegeman, and proven empirically in 2024 by Mucsanyi et al. If the reviewer feels very strongly about this, we will add a paragraph to further clarify it.

---

### Author Response · Authors · 2024-12-23
**Request for Extension Due to Festive Season**

Dear Area Editor,

We hope this message finds you well!

We are writing to kindly request an extension of 7–10 days for our upcoming deadline. Given the festive season and associated commitments, it has been challenging to dedicate the necessary time and focus to complete the required work by the original due date.

We sincerely appreciate your understanding and flexibility. If you need any further information or documentation, please let us know. Thank you for your consideration, and we look forward to continuing to work towards a high-quality submission.

Warm regards!

---

### Decision · Action_Editor_86DA · 2025-01-21

**Recommendation:** Accept as is

**Comment:**

All three reviewers agree that the paper provides sufficient support for its claims
and is of high enough quality to be relevant to the TMLR readership.
I share their opinion and therefore recommend acceptance.

**Audience:**

Uncertainty quantification is a relevant topic for the broader literature and the TMLR audience.
Given the recent popularity of conformal prediction, I expect its combination with the area
of imprecise probabilistic machine learning to be of interest to the TMLR community.

**Claims And Evidence:**

The authors combine conformal prediction with imprecise probabilistic machine learning.
Their theoretical claims are clearly stated and proven.
All empirical claims are supported by a series of experiments.